# An Interpretative Phenomenological Analysis of the lived experience of sensuality expression among women over 50 years of age in Nigeria

**Olubunmi Adeduntan Lawal** [ID]*, **Sinegugu Evidence Duma** [ID]

Discipline of Nursing, School of Nursing & Public Health, College of Health Sciences, University of Kwazulu-Natal, Durban, South Africa

* olubunmilawal09@gmail.com

## Abstract

### Introduction

Sensuality, an essential component of sexuality, is the enjoyment, expression, or pursuit of physical and sexual pleasure or satisfaction. Sensuality expression of women over 50 is under-researched and often ignored, making it difficult to have a scientific basis to develop age-appropriate healthy-ageing programmes for this group in Nigeria. An exploratory study was conducted to explore the lived experiences of the expression of sensuality of Nigerian women over 50 and the meaning they attach thereto.

### Methodology

An Interpretative Phenomenological Analysis approach was used to collect and analyze data from 17 female teachers from three public secondary schools in Osun state, Nigeria, to represent a homogenous group of professional women over the age of 50. A semi-structured questionnaire was used to obtain qualitative data that was thematically analyzed.

### Findings

Four superordinate themes emerged: 'Self-reinvention to camouflage ageing realities for sensuality expression'; 'Embracing own sensuality'; 'Yearning for old self'; and 'Loss of interest in romantic relationships', with various subordinate themes.

### Conclusion

These finding provide the basis to develop age-appropriate healthy-ageing programmes for this group, and a baseline for further sexual health research among this group of women in Nigeria, who are often overlooked or considered asexual due to their being beyond the reproductive age.

**Data Availability Statement:** All relevant data are within the paper and its Supporting Information files.

**Funding:** The author(s) received no specific funding for this work.

**Competing interests:** The authors have declared that no competing interests exist.

## Introduction

Sensuality, is the awareness, acceptance, and enjoyment of one's own body and that of others, especially that of a sexual partner and it includes positive feelings about body image, experiencing eroticism or pleasure, and feeling physical attraction for another person [1]. Sensuality is a component of sexuality, as described in the Circles of Sexuality Model developed by Dennis Dailey in 1981, which uses 5 interconnecting circles to present a holistic approach to human sexuality. Each circle represents a vital part of human sexuality, these being: Sensuality, Sexual intimacy, Sexual identity, Reproduction, and Sexual health and Sexualization. Each circle consists of several components that describe the characteristics of the circle and their influence on human sexuality, and can interact with and affect any or all of the others [2].This study, however, is concerned with the first circle of sexuality which deals with sensuality.

The subject of sexuality of women above 50 is often overlooked, neglected and unspoken [3, 4], despite this being a lifelong expression that plays an important role in graceful ageing due to its physiological, psychological and emotional benefits, its contribution to a person's dignity and social life, and a direct link to increased quality of life [5]. This makes it difficult for people to access accurate information on issues relating to the expression of their sensuality and to receive the support and counselling services necessary for their overall well-being [6, 7].

Women in this age group are sometimes perceived as transiting from 'visibility to invisibility' as their beauty and sex appeal, especially to men, begin to fade, and they are commonly assumed to have become asexual [8]. This is succinctly captured by Marge Piercy in her (2006) poem titled "I met a woman who wasn't there", which explains the feelings commonly experienced by women in their later-life. She observed that in cultures where beauty and femininity are measured by youthfulness, there comes a period in the life of women, usually around the age of 50, when they become unnoticeable and no longer attract the attention of men and younger women, no matter how healthy, well-groomed and nicely dressed they are [9].

In addition, the women may sometimes perceive themselves as being no longer attractive to others [10], their changing bodies being seen as a contrast to beauty and desirability [11]. In many societies, Nigeria inclusive, the expression of sensuality by women over 50 is regarded as either non-existent, unacceptable, or a subject of humour [11, 12]. At this stage of life, even women who had been praised as beautiful in their youth suddenly realize that no one is looking at them anymore. This situation is specifically the case for African women over 50 years globally, as their sexuality is one of the 'unspeakable things unspoken' [13, 14], some having described this period as 'void' or 'empty' [14].

As they age, many women become dissatisfied with their ageing bodies and facial attractiveness, which are becoming less valued, and strive to attain an ideal body weight that is dictated by society. This has bearing on the socialization of the women themselves where they also accept this stereotype and act accordingly [15], specifically for most African women over 50 in African countries, including Nigeria where the study was conducted.

They hold the traditional belief that they are no longer seen as youthful and attractive by men, no matter how hard they try, and that they should take on the role of the elderly grandmother once they are 50 years and over. This belief by the women affects their perspectives on how they see themselves, and may result in their having a negative self-image [16].

Regarding sensuality expression, which is an essential component of sexuality, society often defines the acceptable standards of behaviour for elderly women, thus creating the social image and stereotype that they should be asexual. The same society permits men of the same age to act their age, but regards them as being more attractive as they age [16]. There is evidence that as most women age, they stop regarding themselves as attractive, but become preoccupied with supposedly more important social rather than sexual roles. They tend to change

their self-image to asexual beings in conformity with societal expectations, once they become mothers and grandmothers [17], which can negatively affect their expression of sensuality.

Researchers have shown little or no interest in studying the sexuality expression of women over 50 in sub-Saharan Africa despite sensuality being considered an important aspect of their quality of life [18]. Conducting research that promotes understanding of how women above 50 experience and express their sensuality in all its dimensions is important to promote their holistic health. This is the case in Nigeria where researchers have sidelined women over 50 as sexual beings [18]. This study therefore explored the lived experiences of the expression of sensuality, and the meaning they attach thereto, amongst Nigerian women over 50. The study projects the voice of this group of women, thereby providing knowledge and breaking the silence about the peculiar sensuality needs of this group of women, and their perceived invisibility in the society. Not only will it fill a research gap, but it will also provide a baseline platform from which to conduct other research and develop strategies that enable these women, and society, to change the way they are perceived in relation to the expression of their sensuality without prejudice or bias.

## Materials and methods

### Research design

An Interpretative Phenomenological Analysis (IPA) approach was used due to its ability to provide a deeper understanding and rich descriptions of the expression of sensuality by women over 50 and its commitment to exploring how people make sense of their major life experiences [19]. IPA is useful as a methodology for studies that are complex, ambiguous, and emotional in nature and that involve a homogenous group [20].

IPA as a methodology is most suitable for this study because of the complexity of sexuality and its expression especially among women over fifty years, and the meanings they attach to such experiences. The IPA is also able to examine human experience in the way that it occurs and in its own perspective [19].

IPA researchers view human beings as 'sense-making creatures', and therefore believe that the description of a phenomenon by a participant will reflect their attempt at making sense of their experiences. Furthermore, IPA is idiographic, that is, it is committed to the detailed examination of each case or experience, committed to knowing in detail a person's particular experience and what sense the person is making of his or her experience, before moving on to another [19]. These features set IPA apart from other phenomenological approaches.

Data analysis in this study followed very closely the steps recommended by Smith, et al. [19] as follows:

*STEP 1* –**Reading and Re- reading**

**STEP 2** –**Initial Noting**

**STEP 3** –**Developing Emergent Themes**

**STEP 4** –**Searching for Connections across Emergent Themes**

**STEP 5** –**Moving to the next case**

**STEP 6** –**Looking for patterns across cases**

### Study setting

The study was conducted in three purposively selected public secondary schools in the towns of Ile-Ife, Ilesa, and Osogbo Osun State, South-West, Nigeria, being approximately 50–60 km

from each other, the inhabitants being predominantly Yoruba. The state had an estimated population of approximately 4.7 million people in 2016, of whom females accounted for 1.7 m (36.17%), those aged 50 to 59 years being estimated at 186,745 (10.98%) [21]

## Sampling and sample size

The target population for this study was composed of female teachers aged 50 years and older who worked in public secondary schools in Osun State, Nigeria. They were selected as they are an homogenous group of professionals who are easily identifiable, accessible, and are respected in the community. Purposive sampling was used to select participants; this is also called judgment sampling due to the researchers' making a judgement about which participants to choose based on their qualities and the rich information they can provide on the phenomenon being studied [22]. However, to avoid bias, the participants were encouraged to freely give their consent to participate in the study, following adequate information about their expectations. To be eligible to participate in the study, the women had to be 50 years and over, employed as a teacher in a public secondary school in Osun State, Nigeria, able and willing to communicate their thoughts, feelings and experiences related to the expression of their sexuality and voluntarily consent to participate in the study.

This sample was determined through data saturation, this being the point in the data collection and analysis process when no new themes emerge and a point of redundancy is reached [23]. This sample size of 17 was regarded as adequate as IPA studies are conducted on small sample sizes, where the purpose of the study is to present the detailed perceptions and understandings of the participants undergoing the experience, rather than making general assertions [19].

## Participant recruitment

Permission to access the teachers was received from the Special Education Department of the Ministry of Education, Osogbo, Osun-state, Nigeria and the identified schools. An initial meeting was held with the principals of the three schools to obtain permission to conduct the study in their schools, after which a meeting was held with all female teachers in the schools to introduce the topic and outline the research eligibility criteria for participation. Of those who met the inclusion criteria, 17 of those who showed interest were recruited to participate.

Each potential participant was given the information sheet about the study and requested to sign informed consent. The dates and times for the interview sessions were scheduled at the venue and time of their preference, with most preferring it to be held in their private offices after schools closed in the afternoon

## Data collection methods

A pilot study was conducted in February 2019 with two female teachers over 50 years in one public secondary School in Ile-Ife, Osun State, Nigeria. This provided the opportunity to test the clarity of the questions of the semi-structured interview guide, and identify and prevent any risks associated with the research activity [24]. The pilot data were included in the analysis of the main study as the instrument did not change substantially, and there is no potential risk of contamination that could compromise the quality of the findings [25]. This is allowed in qualitative studies where there is no risk of data contamination. The potential risk identified was possible reluctance to disclose sensitive matters during face-to-face interviews. This was prevented in the main study by assuring participants of confidentiality, and conducting the interviews in safe and supportive environment of the participant's choice

Qualitative data were collected from March to December 2019 through in-depth face-to-face individual interviews using a semi-structured interview guide designed by the researchers in line with the research objectives which focused on the sensuality component of the 5 Circles of Sexuality. This area relates to the use of the senses to experience physical and psychological pleasure, including issues such as body image, aural-visual stimuli, skin hunger, sexual response cycle and fantasy.

The semi-structured interview guide consisted of the following questions. Since you turned 50: "What physical changes have you experienced in your body?" "How do these changes make you feel?" "How have you expressed these changes to others, including your partner, male colleagues and younger females?""How do others respond when you express yourself with your new body changes?" "How do their reactions make you feel?" "How do you perceive yourself since you turned 50?""Do you feel attractive to others?" "How is your relationship with your husband?" "Are you attracted to other men?" "What are you doing to remain attractive and camouflage your ageing looks?". Further probing questions were asked for clarifications as necessary. There is no perceived stigma attached to these questions

The interviews were conducted in English, which is an official language in Nigeria [26], and helped to avoid the difficulties that might result from the variations in intonations and dialects of the Yorubas found amongst teachers in the participating public secondary schools. All the interviews were conducted between 2.30 pm and 4 pm on week days after the close of school, lasted for 60–90 minutes, and were audio recorded with their permission, with the researcher taking field notes to record nonverbal cues and gestures to ensure that all information was captured and were correctly interpreted.

## Data analysis

Each participant was assigned a pseudonym to ensure confidentiality, their verbatim transcript being produced within 24 hours of each interview session while the information was still fresh in the researchers' mind and to start the initial process of data analysis. Data were manually analyzed following the six steps of IPA data analysis as described by Smith et al. [19], and ran concurrently with data collection in an iterative manner. The data from each interview underwent preliminary thematic analysis before the next session was conducted. This iterative process of data collection and data analysis continued until no new information was received (data saturation), which was after 17 participants [27].

Data analysis started with the researchers getting immersed in the data by reading and rereading the transcripts and the field notes while relistening to the recording to gain an understanding of the participant's account and how certain sections of the interview fitted together. Specific ways in which the participant talked, understood, and thought about issues were identified by making notes on descriptive, conceptual, and linguistic comments of interest on the left-hand margin of the transcript. Thereafter, common themes emerging from their data were identified and written on the right margin on the transcript. The emerging themes (which in recent times are being referred to as experiential statements) were organized into a structure while forming connections, patterns, and interrelationships between comments made against the transcripts. Each comment was then grouped in a logical order and assigned a theme name, with similar themes being put together and a new name given for the cluster. These steps were repeated for each transcript while treating each new case independently (Case by case analysis). The themes that developed from all participants' transcripts were examined to determine how they illuminated each other (Cross-case analysis) and to identify similarities among them. Finally, data with similar meanings were grouped into the various categories to produce four superordinate and nine subordinate themes.

## Ethical considerations

The study was guided by the ethical principles of the Declaration of Helsinki, which include the codes of ethics for human subjects [28]. Ethical clearance was obtained from the Biomedical Research Committee (BREC) of the University of KwaZulu-Natal, Durban, South Africa (Ref: BE 490/18), and Obafemi Awolowo University Teaching Hospitals Complex, Ile-Ife, Nigeria Ethics and Research Committee, (Ref: ERC/2018/11/07). Written gatekeeper's permission was obtained from the Schools and Special Education Department of the Ministry of Education, Osogbo, Osun State, Nigeria.

Written informed consent was obtained from all participants who voluntarily agreed to participate after receiving the information sheet and a verbal explanation of the study purpose, benefits and risks. Each participant was given a duplicate copy of the information sheet with the researchers and supervisor's contacts for any enquiry if necessary. A referral system to a Counsellor was established for any participant who experienced any psychological distress during the interview as a result of describing their lived experiences.

## Trustworthiness

Trustworthiness and rigor of the study were assessed using the standard criteria of credibility, confirmability, dependability, and transferability [29–31]. For credibility which is internal validity and confidence in the truth of the findings, member checking was done with 13 of the participants to cross-check and validate the data, ensure accuracy of data collection, and confirm the researcher's interpretations by the participants as reflecting the meanings of their experiences. Two of the first author's peers who served as intercoder assisted with comparing the themes developed by the authors to ensure correct analysis. The second author, an experienced qualitative researcher, supervised and audited the data collection and analysis process in line with IPA guidelines and revised the themes that were developed [19].

Confirmability (objectivity) which is a degree of neutrality or the extent to which the findings of a study are shaped by the participants and not the researcher's bias, motivation, or interest [30, 32] was achieved by differentiating between researcher's values and those of the participants using reflexivity. Dependability which relates to reliability was achieved through consistent and accurate implementation and documentation of the research methods and processes. The interview guide was used to ask all participants the same questions and ensure that the findings were consistent and could be repeated [30]. Transferability, a form of external validity or generalizability, was ensured by providing thick descriptions of the sampling technique and data collection procedures to guide other researchers who might want to replicate the study [33].

## Findings

The ages of the 17 participants ranged from 50 to 58 years, of whom 12 were married and 16 were university first-degree holders, while one had a diploma certificate in education, both qualifications being acceptable for teachers (Table 1).

Four themes and eight sub-themes were generated from the findings (Table 2).

## Theme 1: Self-reinvention to camouflage ageing realities for sensuality expression

Self-reinvention means identifying patterns, values or activities that are no longer necessary or appropriate and changing them [34]. This theme was developed from data about their expression of sensuality in terms of their attractiveness to others and what they are doing to maintain their attractiveness and camouflage their ageing looks. The women indicated that this entailed

**Table 1. Demographic characteristics (n = 17).**

| S/N | Pseudonym | Age (Years) | Education | Marital Status |
|---|---|---|---|---|
| 1. | Toyin | 53 | First degree | Married |
| 2. | Nike | 56 | First degree | Married |
| 3. | Abimbola | 52 | First degree | Married |
| 4. | Bunmi | 53 | First degree | Married |
| 5. | Busayo | 50 | First degree | Divorced |
| 6. | Funmi | 56 | National Certificate in Education | Married |
| 7. | Yemisi | 54 | First degree | Married |
| 8. | Kofo | 55 | First degree | Married |
| 9. | Tolu | 54 | First degree | Married |
| 10. | Kikelomo | 58 | First degree | Widowed |
| 11. | Anike | 55 | First degree | Widowed |
| 12. | Tola | 50 | First degree | Married |
| 13. | Lolade | 53 | First degree | Married |
| 14. | Odun | 54 | First degree | Married |
| 15. | Bolatito | 56 | First degree | Separated |
| 16. | Eunice | 58 | First degree | Widowed |
| 17. | Olusola | 57 | First degree | Married |

embracing new things to hide or disguise the appearance of physical attributes that hamper the enjoyment or pursuit of physical or sexual pleasure which had occurred due to their age. The three sub-themes 'reinventions of self to enhance youthful looks': 'reinventions of body physique' and 'reinventions of self for health reasons' were developed.

**Sub-theme 1. Reinventions of self to enhance youthful looks.** This sub-theme is related to their thoughts, feelings, and attitudes on the physical changes in their bodies. Some felt that the changes in their hair due to ageing made them less attractive and portrays them as old, which negatively affected their expression of sensuality as narrated in the following quotes:

*"My hairline has receded, and my hair is now fluffy and sparse. But that is not a problem. I now own a variety of wigs: long and short, curly or wavy. I wear the different styles of wig to project the image of someone with a full head of hair, which is classy and elegant".* **(Nike, 56, married)**

*"My hair is turning white and becoming thin, which makes me look older than my age. I plan to start applying hair dye to return it to my younger days look when I used to have dark full*

**Table 2. Themes and sub-themes of their experience of sensuality expression and attached meaning.**

| Themes | Sub-Themes |
|---|---|
| 1. Self-reinvention to camouflage ageing realities for sensuality expression | 1. Reinventions of self to enhance youthful looks<br>2. Reinventions of body physique<br>3. Reinventions of self for health reasons |
| 2. Embracing own sensuality | 1. Self-acceptance<br>2. Happiness |
| 3. Yearning for old self | 1.Yearning for youthful body and attractiveness<br>2.Yearning for a trim figure for attractiveness |
| 4. Loss of interest in romantic relationships | 1. Self-imposed romance-less life |

*hair. We all know that grey hairs portray one as old. I do not want that image yet".* **(Kofo, 55, married)**

*"My hair has become thin, sparse with strands of grey hair. I can no longer style my hair like I used to do. So now, I must be creative in my hairstyle or sometimes I wear a wig. I am beginning to think of cutting it low. That way, I will not have to worry about styling it anymore. Luckily, low-cut for ladies is coming back in vogue".* **(Odun, 54, married)**

Some participants narrated the experience of their sagging or droopy breasts that looked like those of an old woman, which they disliked, as it affected their youthful look. Reinventing self was therefore important to make the breasts pointed like that of a young woman for appropriate sensuality expression as indicated by some of the women:

*"I do not like the way my breasts dropdown. I am not fat like most women my age with big breasts, but my sagging breasts spoil the show, and this reduces my attractiveness. I wish my breasts would remain pointed and firm, so, I wear firmer bra".* **(Toyin, 53, married)**

*"I try to package my sagging breasts well in a good, pointed bra (laughs). Well-padded breasts will make my dresses fit more and make me look more attractive".* **(Kofo, 55, married)**

*"I try to support my sagging breasts with a padded bra to make it look bigger and upright. This will bring out my beauty and make me more attractive like I used to be".* **(Toyin, 53, married)**

Other things narrated by the women to accentuate their youthful looks included using expensive creams to have a fresher youthful face and creating time to pay more attention to themselves:

*"My face is not as fresh as before, I can see some areas of wrinkles already, but I started using an expensive cream to smoothen the wrinkles and give my face a glow".* **(Lolade, 53, married)**

*"Now that my children are grown up, I have more time to pay attention to my looks, to remain attractive and appear youthful still. Men still compliment my looks when I try to look good".* **(Tola, 50, married)**

**Sub-theme 2. Reinventions of body physique.** Most of the women experience weight gain as they age and a loss of skin elasticity which they dislike, as they believe it reduces their attractiveness and makes them less valued. Some participants indicated the reinventions they used to keep their body physiques in good shape and condition as part of their sensual expression:

*"It is strange, looking into the mirror, and I could hardly recognize the creature that I saw there. She has a double chin, loose, sagging skin around the face, sagging breasts, extra fat around the middle that was not there before. I see an older woman, but I am working on getting my body back in shape to renew my youthful physique".* **(Olusola, 57, married)**

*"I disliked getting fat. I always pay attention to my physical appearance and more importantly, what I consume; I put in effort to remain trim and not to gain excess weight. Knowing that a trim figure preserves my youthful look gives me satisfaction and a sense of accomplishment".* **(Yemisi, 54, married)**

*"I am now finding it difficult to maintain an average weight for my age. I must monitor my weight more closely now unlike before. I would not like to lose my shape at this age, so I eat less and exercise more".* **(Tola, 50, married)**

*"Initially when I started gaining weight, I was very fat and unattractive, but I have worked on shedding the excess weight. I deliberately reduced my portion, and I eat light meals at night. I worked hard to achieve and maintain my ideal weight. I was avoiding junks, eating well, and exercising. I did all these to maintain my slim figure and preserve physique".* **(Kikelomo, 58, widowed)**

**Sub-theme 3. Reinventions of self for health reasons.** Some participants narrated making conscious efforts to change certain situations in their daily living to prevent or correct some age-related health issues, such as painful joints, high cholesterol levels, increased blood pressure and blood sugar as noted in the following quotes:

*"At some time after I turned fifty, I was feeling pains in the bones and joints in my legs, it became difficult to rise from sleeping or sitting positions. I got my relief after I joined a health fitness club. Now, I am as fit as a fiddle".* **(Bunmi, 53, married)**

*"After I turned 50, I realized I had become very fat and unattractive, and my blood sugar level became higher than normal. To correct this, I started working on shedding the extra weight and controlling my blood sugar level by watching what I eat and taking 30 minutes walk every morning. I need to look good for myself and health reasons".* **(Eunice, 58, widowed)**

*"I was getting easily tired after little activity. When I realized that this is not good for my health. I started to exercise in form of walking for 30 minutes every day"* **(Tolu, 54, married)**

*"At some time after I turned 50, I was obese and my blood pressure was high for my age. I had to go on a special diet and exercise programme for weight reduction. Now, my weight and my blood pressure are fine".* **(Kikelomo, 58, widowed)**

## Theme 2: Embracing own sensuality

This superordinate theme which relates to the meaning the women attached to their sensuality expressions over the age of 50 has two subordinate themes of self-acceptance and happiness.

**Sub-theme 1. Self-acceptance.** Participants highlighted embracing every part of their attributes, positive or negative, as they age as part of their sensuality expression. To affirm this self-acceptance, they indicated that they had started to do the things that they find pleasing without trying to impress others, as depicted in the following quotes:

*"I am not as attractive as before. I feel I am getting old. My skin is no longer as spotless as before, may be because I stopped using the expensive body cream I was using. I used to pay so much attention to my body to be attractive to men, not necessarily my husband alone, but men in general. But now I see no need to impress anyone, so why bother?"* **(Nike, 56, married)**

*"There was a time I was using a particular cream to lighten the dark skin of my face and to smoothen the wrinkles on my face and neck to maintain my youthful look. Everyone I know including my family, remarked how toned my skin was, and I was happy they noticed, but now I stopped using that expensive cream. I have come to accept the change, and this is who I am now, they must accept me for who I am".* **(Funmi, 56, married)**

*"I did not have much time and resources to pay attention to my physique when my children were younger. Caring for them took up much of my time and resources. Generally, I can say that I am prettier now that I am over 50 because the children demand less attention, and they can take care of themselves to a large extent. Now, I have more time and resources to pamper myself and look attractive. I like the new me".* **(Kofo, 55, married)**

**Sub-theme 2. Happiness.** Some participants expressed a feeling of contentment not only regarding their physical appearance but generally over their lived experiences since they turned 50, and were happy with the way they had aged.

*"I am happy with my body and my looks. I became conscious of my looks when people started remarking how youthful I look for my age. I love the compliments I get from people especially men, regarding my looks. I would say that I love the way I look for my age. Life is good. I feel blessed".* **(Bolatito, 56, separated)**

*"I have lots of experiences that cannot be bought with money. I think I have earned all my scars at 57. I am happy".* **(Olusola, 57, married)**

*"When people see me, they do not believe that I am up to 50. Friends remark that I have not changed much since I was a youth, that I have not changed in stature. I love the compliments I get from people regarding my looks. I would say that I love the way I am. Thank God for that".* **(Busayo, 50, divorced)**

*"I may not be as attractive as I used to be in my twenties or thirties, but I am still attractive for a woman my age. That gives me joy"* **(Lolade, 53, married)**

## Theme 3: Yearning for old self

This theme emerged from the meaning the participants attached to their lived experiences of the reactions from men and youth over their physical appearance. Some expressed feelings of longing for the past when their bodies could be used for sensuality expressions and being attractive to men. They reminisced about how attractive they were and compared that to who they think they have become now that they are above 50, and who they would want to be in the near future. Two sub-themes emerged, namely: 'yearning for youthful beauty and attractiveness' and 'yearning for a trim figure for attractiveness'.

**Sub-theme 1. Yearning for youthful beauty and attractiveness.** Some participants indicated experiencing some dissatisfaction with the way they currently look and wanted to remain youthful and beautiful for their sensuality expression. The dissatisfaction came from not wanting to see themselves looking like old women. Some even wondered if others find them attractive as noted in the following quotes:

*"When I look in the mirror, I wish that the face staring at me was that of a young and beautiful woman, not this older-looking face. Often, when I am powdering my face, I can see a difference from how I used to look in my younger days. My reflection reminds me that ageing is catching up with me. Makes me wonder what my partner sees when he looks at this face. How I wish I had my pretty looks back so that my partner would admire my face and give me compliments as he used to".* **(Bunmi, 53, married)**

*"I discovered that I have started having wrinkles on my neck and face. I am not comfortable with the changes in my body. I still wish I would always remain attractive as I was when I was*

*younger. To achieve this, I use wrinkle-repair creams and fill-ins to mask the wrinkles".* **(Abimbola, 52, married)**

*"I have noticed some wrinkles on my face and neck, it is a reminder that I am no longer young, that I am growing old. We all wish to grow old, yet have a beautiful body that attracts men, don't we all? I know I do".* **(Nike, 56, married)**

*"I have started to pay more attention to my skin, my face, and my looks generally than I normally would because I wish to remain beautiful and attractive still as I was when I was younger"* **(Tola, 50, married)**

Some of the participants expressed dissatisfaction with their physical appearance and are working hard on their bodies to achieve beauty and attractiveness as described in the following quotes:

*"Imagine waking up one day, looking at yourself in the mirror and you see this creature you could hardly recognize. She has a double chin, loose, sagging skin around the face, sagging breasts, and extra fat around the middle that was not there before. The skin turns dry, cellulite spreading all over, veins popping up on the legs, and worst of all, there's difficulty remembering simple things. However, I am working hard to get my body back in shape to renew my youthful look through diet and exercises".* **(Olusola, 57, married).**

*"How I wish that my breasts will remain firm and pointed; instead of sagging the way they do now. It takes creativity to package them well in a good, pointed bra. How I wish ageing would not have come with all these changes in my body, I wish I can have my old body back".* **(Kofo, 55, married)**

**Sub-theme 2. Yearning for a trim figure for attractiveness.** Some of the participants described being uncomfortable with the changes in their bodies, with most having put on weight and lost their trim figures. They wished to have their shapely and youthful figures back as depicted in some quotes:

*"I discovered that my tummy has become bigger, and I am gaining weight generally. I am no longer the elegant slim lady that I was, I wish I could have my trim figure back and remain attractive like I was in my younger days to be appreciated by others".* **(Abimbola, 52, married)**

*"I am not pleased with the fact that I have grown fat and have become less attractive in the last few years. I admire women my age who still have beautiful shapes. I wish I am like them. I also wish to maintain my pretty figure to remain attractive".* **(Kofo, 55, married)**

*"The extra fat around my waist and abdomen makes me look older than I am. I now engage in abdominal and waist exercises to get my figure back and renew my youthful look".* **(Olusola, 57, married)**

## Theme 4: Loss of interest in romantic relationship

Some of the women, who were still married, indicated that they were no longer interested in romantic relationships, and had self-imposed a romance-less life.

Some participants attached being over 50 to be a time to be less attracted to men as they were in their younger days, and that their time for romance had passed, despite being married. This is described in the following quotes:

*"When I was younger, even inside marriage, there were a few instances when I found other men physically attractive. I sometimes fantasized about having romantic relationships with them. It was fun then, but now, I feel it is unheard of for a woman of my age to feel attracted to men or engage in any form of romance. If you have such feelings, you must kill it immediately".* **(Toyin, 53, married)**

*"I try not to let the changes in my looks bother me. I used to be very romantic when I was younger, even to the point of having other men in my life. So, I stayed in shape to keep their interests. But when I turned 50, it became difficult to keep up. I accepted the fact that men no longer find me attractive because it became obvious that they preferred younger women. So, I stopped dating other men before they sack me. But it was fun while it lasted though".* **(Nike, 56, married)**

*"You know women will always be attractive to men, I believe that men will always find women attractive. Many men still express their feelings, telling me that I am beautiful and things like that. But I no longer respond to such compliments as I used to do when I was younger. My time for such feelings has passed. I have more important things to think about".* **(Abimbola, 52, married)**

*"Though our men and even the youth appreciate women who can remain youthful and slim, without wrinkles and grey hair. They are quick to remind you to "act your age" if you step out of line. So, I would rather respect my age and not show any attraction to men like I would do a few years back".* **(Bolatito, 56, separated)**

## Discussion

This study explored and described the lived experiences of the expression of sensuality among Nigerian women over 50 years old and the meaning they attach to them, specifically regarding how they and others see the changes in their physical appearance which relates to sensuality; the first circle of sexuality. Most of the participants reported a decline in their good looks and how they are re-inventing themselves to enhance or restore their physical attractiveness. Some are not satisfied with their physical appearance and yearned for the bodies they had when they were younger and more attractive. Some are happy with their circumstances or have learnt to accept them, while a few had lost interest in any form of romantic relationship and have imposed upon themselves a romance-less life.

### Theme 1: Self-reinvention to camouflage ageing realities for sensuality expression

Self-reinvention is a process of realigning desires, not necessarily becoming someone else. It involves producing something new from something that already exists, and incorporates all past experiences and lessons into the values and dreams for a better version of self [35]. The media abounds with advertisements and articles on ways by which bodies can be reinvented, which body parts can be improved, remolded, enhanced or transformed [35]. Regarding sub-theme 1: Reinventions of self to enhance youthful looks, the participants upon acknowledging the physical changes that accompanied their advancing age and recognizing the negative effect of these changes on their declining attractiveness, narrated the things they did to cover up the features of ageing. Some felt that their beauty was fading as a result of the realities of ageing brought to the fore by their receding hair lines, scanty, thinning, fluffy and grey hair, and sagging breasts. They described their efforts at self-reinvention in a bid to camouflage these realities of ageing that included dying their grey hair to black, and covering their receding, fluffy

hair with stylish wigs of varying length and styles. Some planned to cut their hair short, as they were of the opinion that a short-cut for older women was back in fashion, while others took to wearing padded bra to make their flabby, droopy breasts more pointed and firmer. Some participants expressed joy and pride in their good looks when they are well-dressed and receive compliments for looking attractive, while some remain negative about their physical appearance.

Regarding sub-theme 2: Reinventions of body physique, the women described their dissatisfaction with excess fat and weight gain resulting in the loss of a trim body shape and attractiveness. They described how they were working hard to achieve their ideal weight and their desired physique, putting in efforts such as dieting and walking for 30 minutes every day to watch their weight and to get it back to what it was when they had beautiful bodies and trim figures, to enhance their youthful look and promote their attractiveness. These are similar to the reinventing actions described by Nimrod, where he refers to self-reinvention as improving, remolding, enhancing, or transforming one's self [36]. Some of the women in this study expressed dissatisfaction with their bodies.

Regarding Sub-theme 3: reinventions of self for health reasons, some participants described their realities of ageing as presenting in the form of physical ailments, such as arthritis, high blood pressure and obesity. To correct these, they undertook weight control measures, such as eating special diets and engaging in regular exercises such as walking for 30 minutes every day, while some got enrolled in fitness clubs. All these they did to regain their lost beauty and attractiveness and for health reasons. The women were able to reinvent self when they embraced their responsibilities and utilized the opportunities that were available to them [37]. This finding is supported by Clarke's study on women experiencing their ageing bodies especially in the detail and extent of the negative scrutiny of their bodies [38]. A similar study by Hurd found that having a healthy, functional body was more valuable to older women than physical appearance [39]. The findings on reinventions for youthful looks and health reasons support the assertions by Ferrari that some women develop a positive outlook on life and realize that they can control their own lives and therefore can engage in reinvention activities to retain their youthfulness and enhance their expression of sensuality [40].

The findings revealed that women over 50 have both positive and negative experiences of their sensuality expression. The negative experiences and meanings are expressed in the reinventing behaviours that they adopt to deal with ageing and related bodily changes, such as the changes in the physique, youthful beauty, and related health interventions. Information in media and advertisements detail the ways in which bodies can be reinvented to whatever is desired, where body parts can be remolded, enhanced or transformed [41].

The findings revealed at least three reasons why the participants in this study reinvented themselves. For instance, in reinventing self to enhance their youthful looks, they reported dying their grey hair to black, using stylish wigs and wearing a padded bra to uplift their breasts. Some even planned to cut their hair short because they believed it was back in fashion for older women to camouflage ageing.

## Theme 2. Embracing own sensuality

The findings further revealed that women over 50 reported meaningful attachments to their experiences of sensuality expression, including self-acceptance and happiness. They have developed a sense of self-acceptance of their bodies regardless of what others thought about their physical attractiveness or lack of it. Similar experiences were reported by Xu, Rodriguez, Zhang, and Liu, whose findings on the meditating effect of self-acceptance in the relationship between mindfulness and peace of mind indicated that some participants expressed

satisfaction with their situations and have found happiness and peace of mind with who they were [42]. The women accepted their bodies as they were, without struggling to please others or living by societal dictates or norms on what an attractive female body should look like, and finding contentment and happiness after the age of 50.

Our findings show that instead of women over 50 years wallowing over self-pity because of their ageing bodies, they were able to embrace themselves thus taking control of the ageing process and expression of their sensuality. This phenomenon of taking control of one's ageing process and expression of their sensuality is supported by Xu et al. [42]. When women take responsibility and ownership of their lives and begin to see their lives from many perspectives, they find it easier to accept themselves unconditionally and are then able to own their sensuality [42].

### Theme 3. Yearning for old self

The yearning for old self for youthful body and attractiveness and a trim figure may somehow sound contradictory to happiness and self-acceptance. However, on deeper analysis, these are not necessarily contradictory because the yearning for youthful body and attractiveness inspired women to engage in the activities described in subthemes of 'Reinventions to enhance youthful looks', 'Reinventions for body physique', and 'Reinventions for health reasons. We thus conclude that yearning for youthful self among women over 50 is similar to the phenomenon as defined by Walters [43].

In their yearning for youthfulness, they bemoaned their ageing bodies, including weight gain, and sagging breasts and how these affect their sensuality expression. Their concerns about their maturing figures were due in part to the prevailing societal stereotypes in most African societies, including Nigeria where the age-related changes in women's bodies are often equated with less attractiveness, often emphasizing the gap between the bodies of older women and those of younger girls in terms of youthfulness and attractiveness [44, 45]. In addition, these women's expression of their sensuality were derived from their lived experiences as well as from social and cultural norms, which labelled women in their fifties and over as "old and undesirable" [45], while they still saw themselves as wanting to be attractive. Their efforts to retain a youthful appearance emphasize the difference in the perceptions between the bodies of older women in terms of youthfulness and how more value is placed on the latter.

Others bemoaned their fate and lamented about their fat, big tummy and having sagging breasts. Despite acknowledging that these features make them less attractive and sometimes unnoticed, they are not willing to do anything about it. Most of the women who feel this way decided that it was not important to be trim and youthful at their age and that it was appropriate to have a romance-less life.

The media and internet have many images of attractive and successful young women with trim bodies. Comparing those to the maturing figures of older women, as well as societal pressure to be young and attractive, often have negative effects on self-concept and self-esteem [46], and can be demoralizing and intimidating [47]. Some women believe that they have no choice in their present situation and are resigned to their fate, while those who have a positive outlook to life realized that they can control their own lives [40].

### Theme 4. Loss of interest in romantic relationship

Our findings also described the experiences of some of the participants who reported no interest in romantic relationships, either because they are estranged from their spouses or were widowed. Despite the notion that the loss of a spouse through estrangement or death often

results in the loss of assistance, companion, passion and intimacy [48], some of the participants decided to live a life without romance.

In most parts of Nigeria, socio-cultural expectations on widowhood practices demand that they remain secluded, in sober mood, and wear demure dark-coloured clothes throughout the mandatory one-year mourning period after the death of a spouse [49]. Similarly, the widows in this study described their lived experiences after the death of a spouse, and indicated observing the mandatory one-year mourning period that precluded romantic relationships. Two widows in the study expressed no interest in romantic relationship and in looking attractive as they had no desire to remarry, and had decided to remain alone, and therefore no one to impress.

Societal pressures often force widows to remain single to avoid stereotypes such as being a husband snatcher or labelled as wayward [50]. This socialization may negatively alter the sensuality of the women, who no longer sees themselves as objects of desire, which affects their attractiveness and their overall quality of life.

There is a dearth of literature on the romantic life of older women who were either unmarried, separated, divorced, or widowed [48]. The findings of this study on loss of romantic interest can therefore be regarded as contributing to this subject. These findings can provide a basis for further research among older Nigerian women on their romantic relationships, and provide insight into the importance of developing awareness campaigns that challenge and address cultural norms that often pressurize and stigmatize unmarried women over 50 to stop dating and remain single [50, 51].

## Strength and limitations

Using IPA to analyze the women over 50's personal experiences and the meanings they gave them provided new insights into the sensuality needs of this age group. It has also highlighted the basis for the development of community-based social and health promotion centers for women over 50, where they can fully enjoy and express their sensuality without fears of being judged by society as non-conforming [5]. By using their voices, the researchers have broken the silence about the sensuality needs of this group and their perceived invisibility in the eyes of the society.

A number of limitations may prevent the results being generalized to a broader group of women in Nigeria, these being that only three schools were included, all from one state, and only 17 women participated in the study. To avoid possible bias that may result from judgement sampling, the participants were encouraged to freely give their consent to participate in the study, following adequate information about their expectations. The choice of a homogenous sample (teachers) and a small sample size (17), which limits the generalizability of the study, is in line with the principles of IPA [52] which were specifically suitable for exploring the lived experiences of women on this sensitive subject [19, 52], and are important in the development of further research questions about this group of women. In order to generate more knowledge in this area of study, we suggest a larger study using quantitative methods so as to reach a larger population of women who would relate their lived experiences of this phenomenon. Further qualitative studies may be undertaken to explore other aspects of sexuality of women in this age bracket for the development of deeper and comprehensive nursing framework to address their sensuality.

Furthermore, this study covered one state out of the six in the Southwest Geopolitical zone of Nigeria. Replicating this study in the other states in the zone will be more representative of women of that culture.

## Conclusions

The study provides insights into sensuality, an essential component of sexuality needs of teachers as professional women over 50 and highlights the basis for developing age-appropriate healthy-ageing interventions for this group. The findings also provide room for the development of further sexual health research among this group of women who are often overlooked or considered asexual due to their being beyond the reproductive age [53]. We recommend research to assist nurses and other multidisciplinary health practitioners to promote an understanding about how women over 50 express all the dimensions of their sensuality, including Nigerian women, about whom very little is known on the topic.

## Supporting information

**S1 File. Transcript for sensuality expression.**
(DOCX)

## Acknowledgments

We are grateful to all the women who participated in this study and for their willingness to share their lived experiences on the sensitive matter of the expression of their sensuality since they turned 50. We are also grateful to the Director, Schools and Special Education Department of the Ministry of Education, Osogbo, Osun State, Nigeria for granting permission to undertake the study. Our gratitude also goes to the principals of the three schools where data were collected for allowing the researchers access to the female teachers and creating an enabling environment for the study, and to all persons who have contributed in one way or the other to the completion of the study.

## Author Contributions

**Conceptualization:** Olubunmi Adeduntan Lawal, Sinegugu Evidence Duma.

**Data curation:** Olubunmi Adeduntan Lawal.

**Formal analysis:** Olubunmi Adeduntan Lawal.

**Investigation:** Olubunmi Adeduntan Lawal.

**Methodology:** Olubunmi Adeduntan Lawal.

**Project administration:** Olubunmi Adeduntan Lawal.

**Resources:** Olubunmi Adeduntan Lawal.

**Supervision:** Sinegugu Evidence Duma.

**Validation:** Sinegugu Evidence Duma.

**Visualization:** Olubunmi Adeduntan Lawal.

**Writing – original draft:** Olubunmi Adeduntan Lawal.

**Writing – review & editing:** Olubunmi Adeduntan Lawal, Sinegugu Evidence Duma.

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
