## [Decision Letter · Decision Letter 0]

31 Jan 2023

PONE-D-22-29372

An interpretative Phenomenological Analysis of the lived experience of Sensuality expression among women over 50 years of age in Nigeria.

PLOS ONE

Dear Dr. Lawal,

Thank you for submitting your manuscript to PLOS ONE. After careful consideration, we feel that it has merit but does not fully meet PLOS ONE’s publication criteria as it currently stands. Therefore, we invite you to submit a revised version of the manuscript that addresses the points raised during the review process.

We look forward to receiving your revised manuscript.

Kind regards,

Grant Rich, Ph.D.

Academic Editor

PLOS ONE

Journal Requirements:

Additional Editor Comments:

Dear author, please read the comments from both reviewers below. The overall recommendation is for Major Revision so revise and resubmit. Please work with a person fluent in English to ensure the writing and grammar is fluent and improved. The article topic is important, and I understand data on such a sensitive issue was hard to collect, and this makes this research especially valuable.

REVIEWER ONE said Minor Revision and wrote:

“The manuscript is technically sound. The authors provide data that supports the conclusions. These conclusions are drawn appropriately based on the data presented. While the authors conducted a pilot study, I would strongly suggest not to include the pilot study data into the analysis of the study. Perhaps the authors might want to explain the choice of doing that.

The sample size is small and the authors acknowledge that. Judgment sampling often does have a possible bias. This may need to be noted in the limitations of the study.

Since sensuality can be a taboo subject, we may never know for certain if the participants were being truthful in the interviews. Could stigma around the topic sway the participants’ responses?

The authors may need to proof edit the document.

I suggest the authors provide a citation to accompany the definition of sensuality. Given the work that the authors have done in this manuscript it may help if they could make more specific recommendations, addressing how this research might help nurses research or how this research helps to fill the research gap. What other research might need to be done to close the gap?”

REVIEWER TWO said MAJOR REVISION and wrote

“First, the topic of sensuality in middle-aged women in Nigeria is interesting. It provides primary data on groups that are minimally studied and adds to the literature. Nigeria is the most populous nation in the African continent.

Comments –

1. Introduction and Study Focus - The authors need to expand on the relevance of this topic and build up a stronger justification for researching this. It is unclear to me the applied value that the researchers aim for. In the abstract, there is a reference to “… the development of age-appropriate healthy-ageing interventions,” but the study is exploratory, and at best, can generate questions for further studies to understand the topic. In addition, in order to hint at the development of “… further sexual health intervention research,” the authors could have included a subgroup of women with sexuality problems in their study.

2. Research Design - This part needs more detail. The IPA methodology for data collection requires further explanation. The primary readers of the study will most likely be those who are versed in qualitative study, but you can expand your readership to include those who may have limited knowledgeable about this type of research design.

3. The manuscript needs editing – see lines 79, 99, 120-122, 157, 509, 598, 625

4. Ln 179 - The Yoruba people in the south western part of Nigeria is an ethnic group with sub-dialects, NOT a sub-tribe. I suggest the authors follow the recommendation of the United Nations and describe groups in Africa in line with how other groups in the global north are described. Different groups in Europe and North America are not referred to as tribes. For an example, nobody refers to people from Wales as the Welch sub-tribe of United Kingdom.

5. References - Why is this reference in upper-case letters (p. 686)?

6. Summary – There is a dearth of research on this focus group and providing primary data on the group, is a plus. I think the manuscript can be strengthened by addressing

Reviewers' comments:

Reviewer's Responses to Questions

**Comments to the Author**

1. Is the manuscript technically sound, and do the data support the conclusions?

Reviewer #1: Yes

Reviewer #2: Partly

2. Has the statistical analysis been performed appropriately and rigorously? 

Reviewer #1: Yes

Reviewer #2: N/A

3. Have the authors made all data underlying the findings in their manuscript fully available?

Reviewer #1: Yes

Reviewer #2: Yes

4. Is the manuscript presented in an intelligible fashion and written in standard English?

Reviewer #1: Yes

Reviewer #2: Yes

5. Review Comments to the Author

Reviewer #1: The manuscript is technically sound. The authors provide data that supports the conclusions. These conclusions are drawn appropriately based on the data presented. While the authors conducted a pilot study, I would strongly suggest not to include the pilot study data into the analysis of the study. Perhaps the authors might want to explain the choice of doing that.

The sample size is small and the authors acknowledge that. Judgment sampling often does have a possible bias. This may need to be noted in the limitations of the study.

Since sensuality can be a taboo subject, we may never know for certain if the participants were being truthful in the interviews. Could stigma around the topic sway the participants’ responses?

The authors may need to proof edit the document.

I suggest the authors provide a citation to accompany the definition of sensuality. Given the work that the authors have done in this manuscript it may help if they could make more specific recommendations, addressing how this research might help nurses research or how this research helps to fill the research gap. What other research might need to be done to close the gap?

Reviewer #2: First, the topic of sensuality in middle-aged women in Nigeria is interesting. It provides primary data on groups that are minimally studied and adds to the literature. Nigeria is the most populous nation in the African continent.

Comments –

1. Introduction and Study Focus - The authors need to expand on the relevance of this topic and build up a stronger justification for researching this. It is unclear to me the applied value that the researchers aim for. In the abstract, there is a reference to “… the development of age-appropriate healthy-ageing interventions,” but the study is exploratory, and at best, can generate questions for further studies to understand the topic. In addition, in order to hint at the development of “… further sexual health intervention research,” the authors could have included a subgroup of women with sexuality problems in their study.

2. Research Design - This part needs more detail. The IPA methodology for data collection requires further explanation. The primary readers of the study will most likely be those who are versed in qualitative study, but you can expand your readership to include those who may have limited knowledgeable about this type of research design.

3. The manuscript needs editing – see lines 79, 99, 120-122, 157, 509, 598, 625

4. Ln 179 - The Yoruba people in the south western part of Nigeria is an ethnic group with sub-dialects, NOT a sub-tribe. I suggest the authors follow the recommendation of the United Nations and describe groups in Africa in line with how other groups in the global north are described. Different groups in Europe and North America are not referred to as tribes. For an example, nobody refers to people from Wales as the Welch sub-tribe of United Kingdom.

5. References - Why is this reference in upper-case letters (p. 686)?

6. Summary – There is a dearth of research on this focus group and providing primary data on the group, is a plus. I think the manuscript can be strengthened by addressing some of the comments above.

6. PLOS authors have the option to publish the peer review history of their article (what does this mean?). If published, this will include your full peer review and any attached files.

Reviewer #1: No

Reviewer #2: No

While revising your submission, please upload your figure files to

---

## [Author Response · Author response to Decision Letter 0]

19 Apr 2023

REVIEWER 1

NO QUERRY RESPONSE

1. While the authors conducted a pilot study, I would strongly suggest not to include the pilot study data into the analysis of the study. Perhaps the authors might want to explain the choice of doing that. The pilot data were included in the analysis of the main study as the instrument did not change substantially, and there is no potential risk of contamination that could compromise the quality of the findings [25]. This is allowed in qualitative studies where there is no risk of data contamination. (Lines 181– 184).

The potential risk identified was possible reluctance to disclose sensitive matters during face-to-face interviews. This was prevented in the main study by assuring participants of confidentiality, and conducting the interviews in safe and supportive environment of the participant’s choice Lines (184 – 187)

2. Judgment sampling often does have a possible bias. This may need to be noted in the limitations of the study. To avoid possible bias, the participants were encouraged to freely give their consent to participate in the study, following adequate information about their expectations. (Line 661 – 663)

3. Since sensuality can be a taboo subject, we may never know for certain if the participants were being truthful in the interviews. Could stigma around the topic sway the participants’ responses?

 Stigma, defined by Oxford Dictionary as “a mark of disgrace associated with a particular circumstance’’ may not be applicable to this phenomenon. Sensuality; an essential component of sexuality, is an integral part of being human. Though it has been established that society may not readily accept the expression of sexuality by women over 50 years, and that the women themselves oftentimes accept this stereotype; but it is not documented that they feel disgraced or stigmatized as a result of expressing their sensuality. Also, in reference to the structured interview questions asked, there were no perceived stigma attached to these questions (Line 194 – 202).

4. The authors may need to proof edit the document. Done

5. I suggest the authors provide a citation to accompany the definition of sensuality. Done (Line 50)

6. Given the work that the authors have done in this manuscript it may help if they could make more specific recommendations, addressing how this research might help nurses research or how this research helps to fill the research gap. What other research might need to be done to close the gap?”

 In order to generate more knowledge in this area of study, we suggest a larger study using quantitative methods so as to reach a larger population of women who would relate their lived experiences of this phenomenon. Further qualitative studies may be undertaken to explore other aspects of sexuality of women in this age bracket for the development of deeper and comprehensive nursing framework to address their sensuality. 

Furthermore, this study covered one state out of the six in the Southwest Geopolitical zone of Nigeria. Replicating this study in the other states in the zone will be more representative of women of that culture (Line 667 – 675).

By using their voices, the researchers have broken the silence about the sensuality needs of this group and their perceived invisibility in the eyes of the society (Line 656 – 658).

REVIEWER 2

1. Introduction and Study Focus - The authors need to expand on the relevance of this topic and build up a stronger justification for researching this. It is well documented that the subject of sexuality of women above 50 is often overlooked, neglected, and unspoken [3, 4], despite this being a lifelong expression that plays an important role in graceful ageing due to its physiological, psychological and emotional benefits, its contribution to a person’s dignity and social life, and a direct link to increased quality of life [5] (Lines 59 – 62).

It is also documented that the stereotype of the sexless older person has continued to influence policy making and research agendas [16] (Lines 91 – 93).

This study projects the voice of this group of women as they relay their lived experiences on the expression of their sensuality, thereby providing knowledge and breaking the silence about the peculiar sensuality needs of this group of women, and their perceived invisibility in the society (Lines 105 -108).

7. It is unclear to me the applied value that the researchers aim for. In the abstract, there is a reference to “… the development of age-appropriate healthy-ageing interventions,” but the study is exploratory, and at best, can generate questions for further studies to understand the topic. In addition, in order to hint at the development of “… further sexual health intervention research,” the authors could have included a subgroup of women with sexuality problems in their study.

 This is not an intervention study. The main focus of this study is to project the voice of this group of women on the expression of their sensuality, thereby breaking the silence about the peculiar sensuality needs of this group of women, and their perceived invisibility in the society.

Larger studies may be undertaken to explore other aspects of sexuality of women in this age bracket for the development of deeper and comprehensive nursing framework to address their sensuality (Lines 668 – 672).

8. Research Design - This part needs more detail. The IPA methodology for data collection requires further explanation Further explanation on IPA as a methodology done (Lines 130 – 137).

9. The manuscript needs editing Done

10. The Yoruba people in the south western part of Nigeria is an ethnic group with sub-dialects, NOT a sub-tribe. Corrected (Lines 204 -205)

11. Reference in upper case Corrected (Lines 725 -726, 801)

---

## [Editor Report · Decision Letter 1]

24 Apr 2023

An interpretative Phenomenological Analysis of the lived experience of Sensuality expression among women over 50 years of age in Nigeria.

PONE-D-22-29372R1

Dear Authors,

We’re pleased to inform you that your manuscript has been judged scientifically suitable for publication and will be formally accepted for publication once it meets all outstanding technical requirements.

Kind regards,

Grant Rich, Ph.D.

Academic Editor

PLOS ONE

Additional Editor Comments (optional):

The authors responded well to the reviewers' comments. The paper makes a significant contribution on an understudied topic. I vote to accept the paper.

Grant J. Rich, PhD LMT BCTMB LSW 

President-Elect Society for Peace, Conflict, and Violence (APA D48)

President-Elect Society for Media Psychology and Technology (APA D46)

Fellow, Association for Psychological Science (APS)

Fellow, American Psychological Association (APA)

Senior Contributing Faculty, Walden University

Juneau, Alaska USA

Dr. Rich's SPN Website: http://rich.socialpsychology.org/

Book Website (Rich, Gielen, & Takooshian, 2017)

http://www.infoagepub.com/products/Internationalizing-the-Teaching-of-Psychology

Book Website (Rich & Sirikantraporn, 2018)

https://rowman.com/ISBN/9781498554831/Human-Strengths-and-Resilience-Cross-Cultural-and-International-Perspectives#

Book Website (Rich, Jaafar, & Barron, 2020) Psychology in Southeast Asia. Routledge.

https://www.routledge.com/Psychology-in-Southeast-Asia-Sociocultural-Clinical-and-Health-Perspectives/Rich-Jaafar-Barron/p/book/9780367492144

Book Website (Rich & Ramkumar, 2022) Psychology in Oceania and the Caribbean, Springer

https://link.springer.com/book/10.1007/978-3-030-87763-7#editorsandaffiliations

Book Website(Rich, Kuriansky, Gielen, & Kaplan, in press)  Psychosocial Experiences and Adjustment of Migrants: Coming to the USA, Elsevier

https://www.elsevier.com/books/psychosocial-experiences-and-adjustment-of-migrants/rich/978-0-12-823794-6

Book 

(Rich, Kumar, & Farley, in contract)* Handbook of*
*Media Psychology* and Technology-The Science and the Practice,* Springer*

---

## [Editor Report · Acceptance letter]

22 May 2023

PONE-D-22-29372R1 

An interpretative Phenomenological Analysis of the lived experience of Sensuality expression among women over 50 years of age in Nigeria 

Dear Dr. Lawal:

I'm pleased to inform you that your manuscript has been deemed suitable for publication in PLOS ONE. Congratulations! Your manuscript is now with our production department. 

Kind regards, 

on behalf of

Dr. Grant Rich 

Academic Editor

PLOS ONE